# Utility of Initial Arterial Blood Gas in Neuromuscular versus Non-Neuromuscular Acute Respiratory Failure in Intensive Care Unit Patients

**DOI:** 10.3390/jcm11164926

**Published:** 2022-08-22

**Authors:** Ahmad R. Abuzinadah, Asma Khaled Almalki, Rinad Zuwaimel Almuteeri, Rahaf Hassan Althalabi, Hanin Abdullah Sahli, Fatima Abdulrahman Hayash, Rahaf Hamed Alrayiqi, Seraj Makkawi, Alaa Maglan, Loujen O. Alamoudi, Noof M. Alamri, Maha H. Alsaati, Aysha A. Alshareef, Sultan Saeed Aljereish, Ahmed K. Bamaga, Faris Alhejaili, Ahmad Abdulaziz Abulaban, Mohammed H. Alanazy

**Affiliations:** 1Neurology Division, Internal Medicine Department, Faculty of Medicine, King Abdulaziz University, Jeddah 21589, Saudi Arabia; 2Neuromuscular Medicine Unit, King Abdulaziz University Hospital, King Abdulaziz University, Jeddah 21589, Saudi Arabia; 3Faculty of Medicine, King Abdulaziz University, Jeddah 21589, Saudi Arabia; 4College of Medicine, King Saud Bin Abdulaziz University for Health Sciences, Jeddah 22384, Saudi Arabia; 5King Abdullah International Medical Research Center, Jeddah 22384, Saudi Arabia; 6Department of Medicine, Ministry of the National Guard-Health Affairs, Jeddah 22384, Saudi Arabia; 7Neurology Division, Department of Medicine, Ministry of the National Guard-Health Affairs, Riyadh 11426, Saudi Arabia; 8Department of Internal Medicine, King Saud University Medical City, College of Medicine, King Saud University, Riyadh 11472, Saudi Arabia; 9Neurology Division, Pediatric Department, King Abdulaziz University Hospital, Faculty of Medicine, King Abdulaziz University, Jeddah 21589, Saudi Arabia; 10Pulmonology Division, Internal Medicine Department, King Abdulaziz University Hospital, Faculty of Medicine, King Abdulaziz University, Jeddah 21589, Saudi Arabia; 11King Abdullah International Medical Research Center, Riyadh 11426, Saudi Arabia; 12College of Medicine, King Saud Bin Abdulaziz University for Health Sciences, Riyadh 11426, Saudi Arabia

**Keywords:** blood gas, respiratory failure, neuromuscular, pulmonary, myasthenia gravis, Guillain–Barré syndrome, amyotrophic lateral sclerosis

## Abstract

**Background:** The arterial blood gas (ABG) parameters of patients admitted to intensive care units (ICUs) with acute neuromuscular respiratory failure (NMRF) and non-NMRF have not been defined or compared in the literature. **Methods:** We retrospectively collected the initial ABG parameters (pH, PaCO_2_, PaO_2_, and HCO_3_) of patients admitted to ICUs with acute respiratory failure. We compared ABG parameter ranges and the prevalence of abnormalities in NMRF versus non-NMRF and its categories, including primary pulmonary disease (PPD) (chronic obstructive pulmonary disease, asthma, and bronchiectasis), pneumonia, and pulmonary edema. **Results:** We included 287 patients (NMRF, *n* = 69; non-NMRF, *n* = 218). The difference between NMRF and non-NMRF included the median (interquartile range (IQR)) of pH (7.39 (7.32–7.43), 7.33 (7.22–7.39), *p* < 0.001), PaO_2_ (86.9 (71.4–123), 79.6 (64.6–99.1) mmHg, *p* = 0.02), and HCO_3_ (24.85 (22.9–27.8), 23.4 (19.4–26.8) mmol/L, *p* = 0.006). We found differences in the median of PaCO_2_ in NMRF (41.5 mmHg) versus PPD (63.3 mmHg), PaO_2_ in NMRF (86.9 mmHg) versus pneumonia (74.3 mmHg), and HCO_3_ in NMRF (24.8 mmol/L) versus pulmonary edema (20.9 mmol/L) (all *p* < 0.01). NMRF compared to non-NMRF patients had a lower frequency of hypercarbia (24.6% versus 39.9%) and hypoxia (33.8% versus 50.5%) (all *p* < 0.05). NMRF compared to PPD patients had lower frequency of combined hypoxia and hypercarbia (13.2% versus 37.8%) but more frequently isolated high bicarbonate (33.8% versus 8.9%) (all *p* < 0.001). **Conclusions:** The ranges of ABG changes in NMRF patients differed from those of non-NMRF patients, with a greater reduction in PaO_2_ in non-NMRF than in NMRF patients. Combined hypoxemia and hypercarbia were most frequent in PPD patients, whereas isolated high bicarbonate was most frequent in NMRF patients.

## 1. Introduction

Acute neuromuscular respiratory failure due to Guillain–Barré syndrome (GBS), myasthenia gravis (MG), or amyotrophic lateral sclerosis (ALS) is a common reason for admission to the intensive care unit (ICU) and ventilation [1,2]. Several reports have noted respiratory failure as an acute initial and isolated presentation of neuromuscular disorders [3,4,5]. Such presentation of isolated respiratory failure in neuromuscular conditions represents a diagnostic challenge at times. More than half of patients who present to ICU with acute neuromuscular respiratory failure lack initial diagnosis at presentation [6]. Hence, there is a need for diagnostic tools to differentiate between patients presenting with neuromuscular respiratory failure (NMRF) and non-neuromuscular respiratory failure (non-NMRF), such as respiratory or cardiac illnesses [6]. Current assessment methods to recognize NMRF involve clinical assessment and a pulmonary function test [1]. However, these methods require patient cooperation, which, in many contexts, is not feasible, with patients presenting to the emergency department due to severe weakness.

Respiratory failure (RF) occurs when the respiratory system fails to maintain gas exchange and is classified into types 1 and 2 according to blood gas (BG) abnormalities. In type 1 (hypoxemic) respiratory failure, the partial pressure of arterial oxygen (PaO_2_) is less than 60 mm of mercury (mmHg), and the partial pressure of arterial carbon dioxide (PaCO_2_) may be either normal or low. Type 1 respiratory failure occurs mainly due to ventilation perfusion mismatch, such as increased dead space in chronic obstructive pulmonary disease (COPD), causing ventilation without perfusion, or due to a shunt whereby the alveoli are perfused but not ventilated, as in pneumonia and pulmonary edema. In type 2 (hypercapnic) respiratory failure, PaCO_2_ is greater than 50 mmHg, and PaO_2_ may be normal or low; this occurs mainly due to hypoventilation [7,8,9]. Patients with NMRF due to MG, GBS, and ALS are thought to present with type 2 respiratory failure [9]. From a practical perspective, no prior study has compared the ABG parameters between NMRF and non-NMRF patients. However, few studies have looked into the utility of BG parameters of patients admitted to ICU who present with respiratory failure and whether BG parameters differ depending on the cause respiratory failure. Early diagnosis of such patients could considerably improve their management and ICU course.

Because the mechanism of respiratory failure differs between NMRF and non-NMRF cases, we hypothesized that the ranges of ABG parameters differ between NMRF and non-NMRF patients. The aim of this study is to define and compare the arterial BG (ABG) parameter ranges and prevalence of abnormalities in NMRF vs. non-NMRF patients. We aim to describe the ABG parameters for different diseases separately with regard to ABG parameter ranges and prevalence of ABG abnormalities.

## 2. Methods

### 2.1. Study Design and Participants

We conducted a retrospective, cross-sectional study between January 2015 and May 2021 at 4 tertiary centers in Saudi Arabia, (King Abdulaziz University Hospital (KAUH) in Jeddah, National Guard Hospital in Riyadh, King Saud University Medical City in Riyadh, and National Guard Hospital in Jeddah). The institutional review boards approved the protocol in each institution. All centers recruited NMRF cases, whereas the non-NMRF cases were recruited from King Abdulaziz University Hospital. We retrospectively collected data for patients who presented with acute respiratory failure (ARF) during the study period. Inclusion criteria included (1) age of 18–80 years; (2) patients admitted to ICU; (3) diagnosis of acute respiratory failure (ARF) as the indication for ICU admission (see below for ARF definition); and (4) patients with ARF due to one of the following diagnoses: (A) GBS; (B) MG; (C) amyotrophic lateral sclerosis; (D) pneumonia; (E) known cases of chronic obstructive pulmonary disease (COPD); (F) known cases of bronchial asthma; (G) known case of Bronchiectasis; (H) heart failure; (I) noncardiac pulmonary edema; and (J) other known causes of ARF, such as pulmonary embolism and pulmonary fibrosis. Our exclusion criteria were (1) intubation due to sepsis, (2) blood gas collected after the date of intubation, (3) cardiac arrest suffered at initial presentation, and (4) coexistent diabetic ketoacidosis (DKA).

### 2.2. Study Groups and Definition of Variables 

Arterial blood gas (ABG) parameters: we refer to the first ABG taken from the patient before or at the date of admission to the ICU. We excluded ABG values that were taken after the date of ventilation. Parameters include pH, partial pressure of carbon dioxide (PaCO_2_) in mmHg, partial pressure of oxygen (PaO_2_) in mmHg, and bicarbonate level (HCO_3_) in mmol/L.

Combined blood gas (CBG) parameters: we refer to the first blood gas taken from the patient (including venous blood gas if ABG was not available) before or at the date of admission to ICU. Parameters included pH, PaCO_2_, PaO_2_, and HCO_3_.

Acute respiratory failure (ARF) refers to cases admitted to ICU due to respiratory distress caused by one of the diseases mentioned in the inclusion criteria (above) with no additional reason for ICU admission. Cases of ARF were identified by chart review, requiring at least two documentations from ICU and/or emergency physicians indicating the presence of ARF or respiratory distress as an indication for ICU admission. We also excluded cases with coexisting indication for ICU admission, such as sepsis (Figure 1: flow chart).

Group 1 was the NMRF group, which included patients with the following:(A)Guillain–Barré syndrome (GBS): Patients presenting with acute-onset generalized weakness and areflexia that reaches the maximum within four weeks; objective evidence of a diagnosis is from an electrodiagnostic test and/or cerebrospinal fluid.(B)Myasthenia gravis (MG): this diagnosis is based on clinical presentation with objective evidence of diagnosis either through positive serology (acetylcholine receptor antibody or anti-muscle-specific kinase antibody) or a positive decrement in response to repetitive nerve stimulation.(C)Amyotrophic lateral sclerosis: indicated by the presence of progressive weakness with upper and lower motor neuron signs and objective evidence of diagnosis in an electrodiagnostic test performed by a neuromuscular specialist.

Group 2 was the non-NMRF group, which included patients with the following:(A)Pneumonia with X-ray or culture confirmation;(B)Known cases of COPD and use of bronchodilators prior to admission;(C)Known cases of asthma and use of bronchodilators prior to admission;(D)Heart failure with objective evidence on X-ray or echocardiogram;(E)Bronchiectasis confirmed by chest CT;(F)Noncardiac pulmonary edema;(G)Others causes of ARF, including pulmonary embolism based on CT angiogram, pulmonary fibrosis, cystic fibrosis, and combined etiologies from the causes mentioned above.

### 2.3. Study Measures

#### 2.3.1. Primary Measures

To define and compare the ranges of arterial blood gas parameters (pH, PaCO_2_, PaO_2_, HCO_3_) in patients presenting with ARF due to NMRF and non-NMRF. 

#### 2.3.2. Secondary Measures

(1)We compared the ranges of ABG parameters between NMRF and the following three categories of non-NMRF:a.PPD (asthma, COPD, and bronchiectasis). This category represents the ventilation perfusion mismatch mechanism;b.Pneumonia: this category represents the acute shunting mechanism; andc.Pulmonary edema: heart failure and non-cardiac pulmonary edema; this category represents the chronic shunting mechanism.(2)We compared the prevalence of acidosis (pH < 7.35), hypercarbia (PaCO_2_ > 50 mmHg), hypoxia (<80 mmHg), and high bicarbonate levels (HCO_3_ > 22 mmol/L) between NMRF and non-NMRF and between NMRF and the three categories of non-NMRF mentioned above.(3)We defined the ranges and prevalence of ABG parameters for each of the diseases included in our criteria separately.(4)We compared the proportion of patients with NMRF and non-NMRF who fulfilled the definition of type II respiratory failure (PaCO_2_ > 50 mmHg).(5)We compared the proportion of patients with NMRF and non-NMRF who had both hypercarbia and hypoxia.(6)We compared the proportion of patients with NMRF and non-NMRF who presented with isolated high bicarbonate levels (defined as >22 mmol/L).(7)We compared the proportion of patients with NMRF and non-NMRF who had either hypercarbia or hypoxia.

### 2.4. Sensitivity Analysis

(1)We defined and compare the ranges of ABG parameters between NMRF and non-NMRF patients in severe ARF cases (defined as requiring intubation for ≥5 days or death due to ARF within 5 days).(2)We defined and compare the ranges of combined BG (CBG) parameters (which includes venous BG when ABG prior to intubation was not available) between NMRF and non-NMRF.(3)We compared data provided by King Abdulaziz University hospital with those provided by other centers for NMRF cases.

### 2.5. Statistical Analysis

The characteristics of patients were analyzed using the median (IQR, interquartile range) and frequencies, as appropriate. Mann–Whitney U tests, χ^2^ tests, and Fisher exact tests were used to compare the data between patients with NMRF and non-NMRF, as appropriate. Because this was a retrospective study, we used a convenient sampling approach, including all available cases that satisfied the inclusion and exclusion criteria; therefore, the sample size was not calculated as priori. Statistical analysis was performed using STATA version 13 (Stata-Corp, College Station, TX, USA).

## 3. Results

The number of patients retrieved from the electronic medical record search was 656. A total of 303 patients with available CBG data were included; of them, 287 patients had ABG data available and were included in the primary analysis (Figure 1, flow chart). A total of 218 patients with non-NMRF and 69 patients with NMRF were included in the study. The former patients were older than those with NMRF; however, the two groups were similar in terms of gender distribution and body mass index (BMI) (Table 1). Additionally, respiratory rates and fractions of inspired oxygen (FiO_2_) were higher in the non-NMRF compared to NMRF group (Table 1).

### 3.1. Primary Measures

The ranges of ABG parameters are shown in Table 2. NMRF patients had higher pH, PaO_2_, and HCO_3_ levels than non-NMRF patients; in contrast, there was no difference in PaCO_2_. We present the ABG parameters data for NMRF and non-NMRF cases in a scatter plot in Appendix A.

### 3.2. Secondary Measures 

The ranges of ABG parameters for each category of the ARF mechanisms are presented in Table 2. NMRF patients presented with less acidosis when compared with non-NMRF patients and when compared with each one of the three major categories of non-NMRF. The highest levels of PaCO_2_ were observed in PPD patients. The lowest levels of PaO_2_ were observed in pneumonia patients. Pulmonary edema and pneumonia patients had the lowest HCO_3_ levels.

**Table 2 jcm-11-04926-t002:** Comparing ABG parameters between NMRF and non-NMRF categories.

	pH	*p* Value *	PaCO_2_	*p* Value *	PaO_2_	*p* Value *	HCO_3_	*p* Value *
Neuromuscular respiratory failure, median (IQR)	7.39 (7.32–7.43)		41.5 (35.3–49.6)		86.9 (71.4–123)		24.8 (22.9–27.8)	
Non-neuromuscular respiratory failure, median (IQR)	7.33 (7.22–7.39)	<0.01	43.9 (35.9–62)	0.13	79.6 (64.6–99.1)	0.02	23.4 (19.4–26.8)	<0.01
Primary pulmonary disease (COPD, asthma, and bronchiectasis)	7.29 (7.21–7.36)	<0.01	63.2 (46–77.3)	<0.01	79.9 (68.7–97.2)	0.12	27.5 (22.8–29.8)	0.33
Pneumonia, median (IQR)	7.355 (7.27–7.4)	0.01	41 (35.6–55.9)	0.84	74.3 (61.1–94.5)	<0.01	23.4 (19.6–25.5)	<0.01
Pulmonary edema (cardiac and non-cardiac), median (IQR)	7.33 (7.21–7.39)	<0.01	42.4 (33.4–53.9)	0.85	86.6 (65.2–101)	0.32	20.9 (17.4–25.5)	<0.01
Others, median (IQR)	7.34 (7.31–7.39)	0.19	49.5 (36.5–64.1)	0.29	83.5 (76.7–101.1)	0.66	24.1 22.5–26.1)	0.48

* *p* value compared to NMRF (neuromuscular respiratory failure), Wilcoxon rank sum (Mann–Whitney) test, IQR: interquartile range.

The prevalence of ABG abnormalities among the various categories of ARF is presented in Figure 2 and Appendix A. Acidosis was less prevalent in NMRF (31.9%) compared to non-NMRF (55.1%) and PPD (67.4%) patients. Hypercarbia was more prevalent in non-NMRF (39.9%) and PPD (67.4%) patients compared to NMRF patients (24.6%) (*p* < 0.05). Hypoxia was more prevalent in non-NMRF (50.5%) and pneumonia (58.7%) patients compared to NMRF patients (33.8%) (*p* < 0.05). High normal or elevated bicarbonate levels were more prevalent in NMRF (76.8%) and PPD (73.9%) patients compared to pulmonary edema cases (44.1%).

The ranges of ABG parameters for each disease are presented in Table 3. The highest levels of PaCO_2_ were observed in bronchiectasis and COPD patients. The lowest levels of PaO_2_ were observed in pneumonia, COPD, and bronchiectasis patients. The highest levels of HCO_3_ were observed in bronchiectasis and COPD patients, followed by ALS. The prevalence of ABG abnormalities for each disease is presented in Appendix A.

We found that 17 (24.6%) and 87 (39.9%) patients with NMRF and non-NMRF, respectively, fulfilled type II respiratory failure criteria (*p* = 0.02). The proportion of patients with NMRF and non-NMRF who had combined hypercarbia and hypoxia was greater among PPD (37.8%) than NMRF (13.2%) patients (*p* < 0.01) (Figure 2 and Appendix A), whereas the proportion of patients who presented with isolated high bicarbonate levels was higher among NMRF (33.8%) compared to non-NMRF (14.8%), PPD (8.9%), pneumonia (15.2%), and pulmonary edema (16.4%) patients (all *p* < 0.01) (Figure 2 and Appendix A). The proportion of patients who presented with either hypoxia or hypercarbia was lower among NMRF (45.6%) compared to non-NMRF (67.1%) patients (*p* = 0.002) and compared to PPD (80%) patients (*p* < 0.001).

### 3.3. Sensitivity Analysis

When analysis was restricted to severe ARF, the results were similar to those of the primary endpoint. The ranges of ABG parameters for each category of the ARF mechanisms are presented in Appendix A. The highest levels of PaCO_2_ were observed in PPD patients. The lowest levels of PaO_2_ were observed in pneumonia patients. Pulmonary edema and pneumonia patients had the lowest HCO_3_ levels. The ranges of CBG parameters are shown in Appendix A (Appendix A); the results were similar to those of the primary endpoints, where NMRF patients higher PaO_2_ and HCO_3_ levels than non-NMRF patients; in contrast, there was no difference in PaCO_2_. A comparison of data between the study centers revealed no differences (Appendix A).

## 4. Discussion

In the present study, we established the ranges of initial ABG parameters in NMRF and non-NMRF patients admitted to intensive care units with acute respiratory failure. We found a lower degree of hypoxemia in patients with NMRF compared to those with non-NMRF. The levels of hypercarbia were similar between NMRF and non-NMRF patients; however, the proportion of patients who had hypercarbia and met the type 2 respiratory failure definition was greater in non-NMRF (39.9%) compared to NMRF (24.6%) patients. Combined hypoxemia and hypercarbia were most characteristic of PPD, whereas isolated high bicarbonate was characteristic of NMRF. A majority of NMRF patients in our cohort presented with either normal or mild hypoxemia and with normal PaCO_2_, which is similar to results reported in a cohort of 79 patients who presented with NMRF in ICU with PaCO_2_ of 48 mmHg and PaO_2_ of 92 mmHg [6]. This is expected, as neuromuscular disorders initially cause microatelectasis, particularly at the lung base, leading to shunting (more alveoli perfused and not ventilated), which causes mild hypoxemia; however, this stage is usually compensated by tachypnea, which leads to hypocarbia and normalization of PaO_2_ [1,2]. This is usually followed by fatigue in the respiratory muscle and leads to hypercarbia, which occurred in 24.6% of NMRF patients in our cohort. In fact, it is suggested that with tachypnea, patients should have hypocarbia, whereas cases of tachypnea with normocarbia indicate ineffective respiration and advanced respiratory failure [1]. The abnormalities in ABG in NMRF cases are due to hypoventilation; there are three factors that influence ventilation: (1) respiratory rate, (2) tidal volume, and (3) dead space (no gas exchange in the airway) [10]. As NMRF reduces the total tidal volume and does not affect dead space, the respiratory rate is the variable that can be modified physiologically to temporarily compensate for the reduced tidal volume in NMRF. Our study showed that patients with NMRF usually (54%) present without hypoxia or hypercarbia during this compensatory phase, at which time ABG parameters could be normal, in contrast with those in non-NMRF patients, who less frequently (32.9%) present without hypoxia or hypercarbia. Both groups (NMRF and non-NMRF) had elevated respiratory rates (RR); however, RR was higher among non-NMRF patients. The median of FiO_2_ was higher among non-NMRF compared to NMRF patients. This, along with the finding that PaO_2_ was higher among NMRF compared to non-NMRF patients, may indicate that with a normal gas diffusion and perfusion mechanism, such as in NMRF, respiratory and ABG parameters are expected to be corrected faster with oxygen supplementations than in diseases that affect gas diffusion and perfusion, such as non-NMRF.

When each neuromuscular disease was analyzed separately, we found that MG and GBS shared a common profile, with a low proportion of patients with hypercarbia (10.3% and 26.1%, respectively) and a low proportion with hypoxia (28.6% and 30.4%, respectively). This is in contrast to the ALS group, where the proportions of hypercarbia and hypoxia were higher (47.1% and 47.1%, respectively) than in the GBS and MG groups. A study examining MG exacerbation revealed PaCO_2_ levels ranging between 28 and 54 mmHg, which is similar to our data; however, the previous study reported a higher level of PaO_2_ than our data, with a mean of 101 mmHg [11]. Regarding GBS, the mean of PaCO_2_ was found to be normocarbic, at 37 mmHg (31–45 mmHg) [12]. Additionally, Kalita et al. found that GBS cases that required intubation had average PaCO_2_ levels of 41 mmHg (37–46 mmHg), which is consistent with our results, whereas average PaO_2_ levels were found to be 70 mmHg (61–93 mmHg) among intubated GBS cases, which is slightly lower than the PaO_2_ levels reported in our cohort [13]. This latter cohort probably included more severe cases, as the authors reported a single breath count of 3 [13]. In GBS and MG cases, hypoxia and hypercarbia occur late, and physicians should not wait for these changes to occur before they provide respiratory support [14]. However, we found that GBS and MG usually have isolated bicarbonate at the higher level. Prior studies have shown that hypercarbia does not develop in ALS unless loss of lung volume becomes severe with a forced vital capacity reduced down to at least 20% of the predicted value [15]. The same authors also found that hypoxia occurs in 50% of the cases that develop hypercarbia. There are no prior data on hypercarbia or hypoxia prevalence among ALS cases admitted to ICU; however, it was reported that at 6 months from ALS onset, 54% of ALS cases have hypercarbia and 27% have hypoxia [15].

Non-NMRF cases span various mechanistic categories. PPD, including COPD, asthma, and bronchiectasis, causes ABG abnormalities through ventilation perfusion mismatch, whereby there is an area in the lung ventilated without gas exchange. Our study showed that the levels and prevalence of hypercarbia are higher in PPD compared to NMRF (63.25 mmHg vs. 41.5 mmHg and 67.4% vs. 24% for PPD vs. NMRF, respectively). Hypoxia occurred more frequently in PPD (51.1%) compared to of NMRF (33.8%), with a median PaO_2_ of 79.9 vs. 86.9 mmHg, respectively. This finding is consistent with studies that have looked at COPD exacerbation cases and found an elevated mean of PaCO_2_ of 59 mmHg [16]. Another recent study looked at COPD cases admitted to ICU with respiratory failure and found the mean PaCO_2_ and PaO_2_ to be 54 mmHg and 63 mmHg, respectively [17]. This is in contrast with another study that showed that most patients with COPD do not have elevated PaCO_2_, whereas PaO_2_ levels vary between 60 and 80 mmHg [18]. Pneumonia and pulmonary edema cause respiratory failure due to shunting, whereby the perfused area in the lung is not ventilated [9]. This usually results in hypoxemia without hypercarbia, which is consistent with our finding. ABG in pneumonia differs from that in NMRF, mainly with a higher prevalence of hypoxemia, whereas the higher prevalence of hypoxemia in pulmonary edema compared to NMRF did not reach statistical significance in our cohort. 

On average, arterial bicarbonate was higher among NMRF cases than non-NMRF cases. When NMRF was compared to different categories of non-NMRF, the difference in bicarbonate was more prominent versus pneumonia and pulmonary edema, whereas PPD had a similar level of bicarbonate. However, an important difference between NMRF and PPD is that patients with NMRF are more likely to present to ICU with isolated high bicarbonate (without hypoxia or hypercarbia) compared to PPD patients. Prior studies that looked at the serum bicarbonate level among patients with ALS have showed that 80% of those with elevated serum bicarbonate died within 5 months, whereas patients with normal serum bicarbonate levels remained alive at 15 months of follow-up [19]. Among MG patients, elevated serum bicarbonate (>30 mg/dL) was found to be a predictor for prolonged intubation [20]. The authors interpreted these results such that high serum bicarbonate levels are probably a more reliable indicator of chronic respiratory acidosis than pre-intubation carbon dioxide partial pressure, which may transiently normalize in some patients with increased respiratory effort. Another study among MG patients reported similar results of more successful noninvasive ventilation if the serum bicarbonate level was <30 mmol/L [21]. 

Our data can be used in conjunction with the clinical context but not as a stand-alone test. The clinical implications of our findings—in the appropriate clinical context—include helping physicians to identify the underlying cause of acute respiratory distress in cases presenting for the first time without a prior clinical history. This will guide further clinical testing and consultation with the relevant specialties to confirm clinical suspicion. In addition, our data may alert physicians to a coexisting disorder contributing to acute respiratory distress in patients known to have certain disease, particularly when the pattern of ABG results does not fit the pattern expected of the known disease. Our data may guide planning of future clinical studies that address respiratory distress in neuromuscular and non-neuromuscular diseases when it comes to using ABG in inclusion criteria or for calculations of effect size or sample size. Nonetheless, we emphasize the importance of taking the whole clinical picture into consideration when interpreting ABG results, rather than considering ABG results in isolation, which could limit the usefulness of our data. Finally, defining the magnitude of ABG changes in NMRF and non-NMRF patients in a single study with similar methodology may contribute to a deeper understanding of the identified concepts associated with acute respiratory failure.

Our study is subject to several limitations. The retrospective and observational nature of the study could contribute inherited bias. We did not incorporate oxygen saturation or respiratory rate data in our analysis. Another limitation of our study is that non-NMRF cases were included from only one center. However, because we collected the initial ABG upon presentation before therapeutic interventions, we believe that our data represent the disease course rather than a variation in the clinical practice in each center, although the later cannot be totally excluded (Figure 1: flow chart). Other measures of pulmonary function, such as vital capacity, were not collected due to difficulties associated with documentation. There were few cases included for some diseases, such as pulmonary embolism and interstitial pulmonary fibrosis. The influence of comorbidities, such as diabetes and hypertension, on the outcomes were not adjusted for. We did not evaluate the response to different treatment modalities, such as prednisone for myasthenic patients and bronchodilator for asthmatic and COPD patients.

In conclusion, our data provide the expected range of ABG changes among patients admitted to ICU, and we found that hypercarbia occurred in a quarter of patients with NMRF. When compared with NMRF, PPD patients presented more frequently with hypercarbia, whereas pneumonia patients presented more often with hypoxia, and pulmonary edema patients have less elevated bicarbonate levels. Combined hypoxemia and hypercarbia are characteristic of PPD, whereas isolated high bicarbonate is characteristic of NMRF.

## Figures and Tables

**Figure 1 jcm-11-04926-f001:**
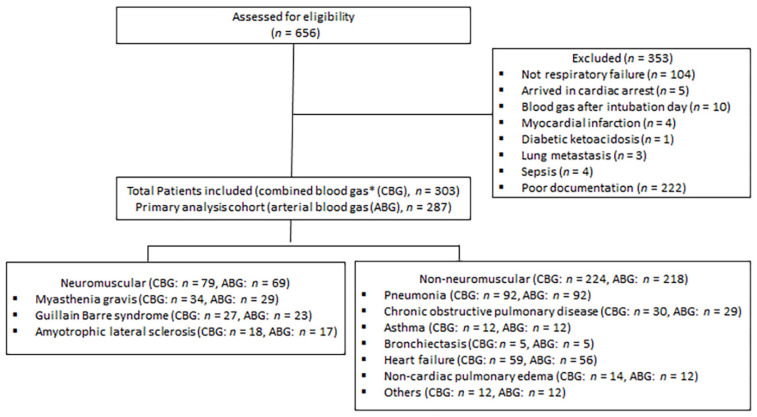
Flow chart as per STROBE (STrengthening the Reporting of OBservational studies in Epidemiology) guidelines. * Combined blood gas includes venous blood gas if the patient did not have arterial blood gas on initial presentation.

**Figure 2 jcm-11-04926-f002:**
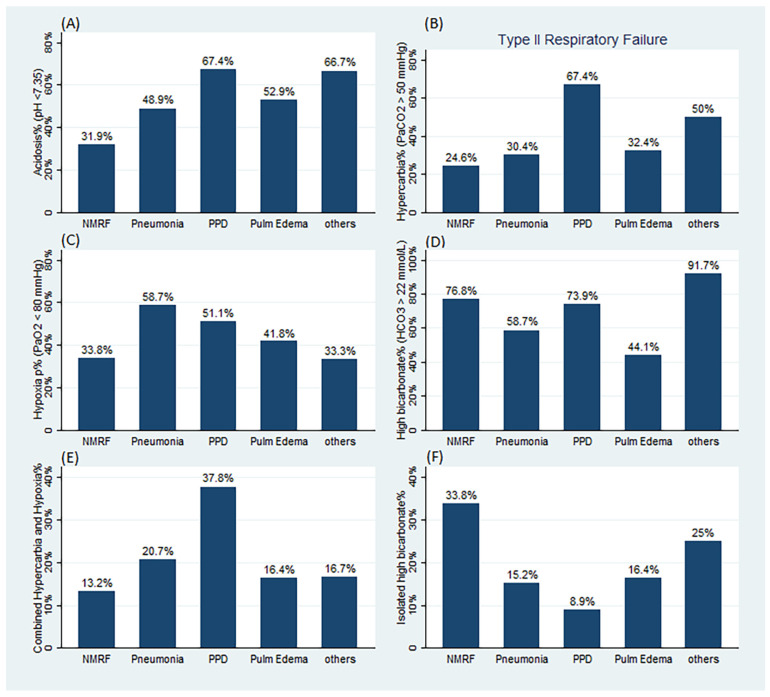
Prevalence of ABG abnormalities per category of respiratory failure. (**A**) Prevalence of acidosis (pH < 7.35). (**B**) Prevalence of hypercarbia and type II respiratory failure (PaCO_2_ > 50 mmHg). (**C**) Prevalence of hypoxia (PaO_2_ < 80 mmHg). (**D**) Prevalence of high bicarbonate level (HCO_3_ > 22 mmol/L). (**E**) Prevalence of combined hypercarbia and hypoxia. (**F**) Prevalence of isolated high bicarbonate level. NMRF: neuromuscular respiratory failure, PPD: primary pulmonary diseases, Pulm Edem: pulmonary edema. For a comparison *p* values, refer to Appendix A.

**Table 1 jcm-11-04926-t001:** Patient characteristics.

	Neuromuscular Respiratory Failure Cases (*n* = 69)	Non-Neuromuscular Respiratory Failure Cases (*n* = 218)	*p* Value *
Age, median (IQR)	51 (34–62)	63 (50–74)	<0.01
Male, *n* (%)	39 (56.5)	120 (55.1)	0.89
BMI, median (IQR)	24.8 (20.5–29)	27 (23.4–31.2)	0.06
Diabetes, *n* (%)	17 (24.6)	123 (56.4)	<0.01
Hypertension, *n* (%)	26 (37.6)	127 (58.2)	<0.01
Cardiac diseases, *n* (%)	6 (8.7)	88 (40.3)	<0.01
Pulmonary diseases, *n* (%)	8 (11.5)	68 (31.1)	<0.01
Creatine kinase level, median (IQR)	79 (43–143.5)	159 (79–375)	<0.01
Days from ABG to ventilation, median (IQR)	0 (0–1)	0 (0–1)	0.12
Respiratory rate, median (IQR)	23 (20–30)	30 (24.5–40)	<0.01
FiO_2_ ^§^, median (IQR)	28% (20–40%)	40% (30–60%)	<0.01
Ventilation needed at any time during ICU stay, *n* (%)	66 (95.6)	209 (95.8)	1.00
Invasive mechanical ventilation needed at any time during ICU stay, *n* (%)	60 (86.9)	137 (62.8)	<0.01
Only non-invasive ventilation needed during ICU stay, *n* (%)	6 (8.7)	72 (33)	
Death, *n* (%)	6 (8.7)	68 (31.1)	<0.01

* Median compared with Mann–Whitney U tests, proportion compared with Fisher exact tests. ^§^ FiO_2_: fraction of inspired oxygen. IQR: interquartile range.

**Table 3 jcm-11-04926-t003:** Arterial blood gas parameters according to disease.

	pH	PaCO_2_	PaO_2_	HCO_3_
Myasthenia gravis, median (IQR)	7.39 (7.32–7.4)	39 (34.5–43.8)	88.1 (76.4–127.4)	23.7 (19.3–26.3)
Guillain–Barré syndrome, median (IQR)	7.39 (7.34–7.43)	42 (37–51)	86.3 (72–106)	25 (23–26.9)
Amyotrophic lateral sclerosis, median (IQR)	7.39 (7.3–7.45)	47.8 (35.3–77.9)	81.1 (58.2–141)	27.8 (24.1–32.5)
Pneumonia, median (IQR)	7.36 (7.27–7.4)	41 (35.6–55.9)	74.3 (61.1–94.5)	23.4 (19.6–25.5)
COPD, median (IQR)	7.29 (7.21–7.36)	67.7 (50.1–78.1)	76 (68.4–90.1)	28.2 (24–29.7)
Asthma, median (IQR)	7.28 (7.21–7.36)	45.8 (36.4–57.4)	89.6 (79.9–128)	20.7 (16.6–23.4)
Bronchiectasis, median (IQR)	7.32 (7.28–7.33)	69 (65.6–76.4)	65 (64.9–83.9)	32.8 (32–36)
Heart failure, median (IQR)	7.34 (7.21–7.39)	42.4 (33.4–55.8)	85.7 (62.8–98.7)	21.25 (17.4–25.6)
Non cardiac pulmonary edema, median (IQR)	7.32 (7.21–7.36)	42.45 (35.3–48.3)	99.15 (81.65–111)	19.55 (17.4–22.7)
Others, median (IQR)	7.34 (7.31–7.39)	49.5 (36.5–64.1)	83.55 (76.7–101.1)	24.2 (22.5–26.1)

IQR: interquartile range.

## Data Availability

All data are available upon request from the corresponding author.

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
