# Peer review of "Utility of Initial Arterial Blood Gas in Neuromuscular versus Non-Neuromuscular Acute Respiratory Failure in Intensive Care Unit Patients"

_jcm, 2022, doi:10.3390/jcm11164926_

Round 1

Reviewer 1 Report

Abuzinadah et a performed a retrospective data collection on patients admitted to the intensive care unit (ICU) with respiratory failure. Patients with respiratory failure due to neuromuscular failure (NMRF) were compared with patients with respiratory failure from other causes (non-NMRF). In total 287 patients were included in this analysis. Abuzinadah et all found patients with NMRF to have a higher pH, higher PaO2, and higher HCO3 compared to patients with non-NMRF. Some additional differences were found when comparing different causes of non-NMRF with NMRF patients. I have some questions and comments. My major concern is the lack of a clear definition of respiratory failure (the main inclusion criteria).

Major comments:

1.    Introduction: the hypothesis is missing, please add.

2. Methods: why were non-NMRF patients included only from King Abdulaziz University Hospital while the NMRF cases came from 4 different hospitals? Medical guidelines and treatments may differ between centers. Because there is a difference in which hospitals the two patient groups were collected from, bias may have been introduced.

3. Methods: a sample size calculation is missing. If no sample size calculation is performed, the methods should state this.

4.   Methods: the inclusion criteria are very vague - and I think this is a major error. Only patients with respiratory failure are included in this analysis (correct?). How is respiratory failure defined? Was an EPD note from clinicians sufficient? Or were certain laboratory results confirming respiratory failure necessary to be eligible for this study? Or was respiratory failure defined as "the need for respiratory support"? Or was every patient with COPD or asthma admitted to the ICU fit for inclusion? Please explain this in more detail. Second, I recommend that the authors reformulate the inclusion criteria. In its current state, the inclusion criteria could be misread/interpreted ("a patient with pneumonia (regardless of whether they had respiratory failure or not) is eligible for this analysis." Try rewording this section (e.g., "patients with respiratory failure due to pneumonia, exacerbation COPD, ..... were included").

5.  A sensitivity analysis on patients with severe respiratory failure was performed. How is severe respiratory failure defined? Please clarify when patients were considered as having severe respiratory failure. This must be documented in the manuscript.

6.    Methods: the authors explain that the ABG was defined as the first ABG taken from the patients before or at the date of admission to the ICU. Some patients may have been hospitalized for many days before ICU admission or required respiratory support. Perhaps respiratory failure developed many days after the initial hospitalization. Is this initial ABG representative of the patient's current condition? How much time was there between the first ABG and the moment the patient was classified as having respiratory failure?

7.  Methods: did the authors correct for different centers in the analysis? Please add centers as random intercept in the analysis or declare why this is not considered necessary.

8.    I am missing the clinical relevance of this paper. How do these findings improve current medicine?

Author Response

Please see the attachment PDF file.

First reviewers’ comments and responses are in pagea 2 to 10 (responses from #1 to #10)

Second reviewers’ comments and responses are in pages 11 to 15 (responses number #11 to #14)

Reviewer 2 Report

The authors present a retrospective review of patients with respiratory failure admitted to 4 tertiary centers between Jan 2015 and May 2021.  

Major critiques

The authors describe two populations of patients with neuromuscular respiratory failure (NMRF) and non-NMRF.  The non-NMRF was recruited from a single center.  The NMRF was recruited from 4 centers.  The authors should explain the reason for this distribution and whether this introduces any potential bias into the dataset.

- The data is presented as a bar graph in Figure 2 for each ABG parameter that averages the data.  A clinician would benefit from a scatter plot representation which might provide insight into how distinct the values are between NMRF and non-NMRF.  The question relates to the distinction between the statistical significance of the findings and the information's clinical value.

- The paper lacks any specific clinical information about the patients other than the arterial blood gas i.e. level of supplemental oxygen or vital signs.

- The authors might consider how this information can be used clinically in their discussion.  Despite the statistical significance of some values, the clinical distinction based on blood gas parameters seems fairly limited.

Author Response

Please see the attachment PDF file.

This is a reply to reviewer 2.

Second reviewers’ comments and responses are in pages 11 to 15 (responses number #11 to #14).

First reviewers’ comments and responses are in pagea 2 to 10 (responses from #1 to #10)

Round 2

Reviewer 1 Report

The authors have answered all review questions and revised the manuscript accordingly. I continue to find the methodology and clinical relevance of this manuscript questionable, but I do not believe this can be further improved. 

Author Response

Thank you very much for the comments. We aknowledged in our manuscript the limitation of retrospective studies as well as the other limitations however, we have spent a lot of time to review the data and methods and to make sure it was done at the highest possible standard in order to get the most accurate results possible. 

Thank you very much